# Individual crop loads provide local control for collective food intake in ant colonies

Efrat Esther Greenwald[†], Lior Baltiansky[†], Ofer Feinerman*

Department of Physics of Complex Systems, Weizmann Institute of Science, Rehovot, Israel

**Abstract** Nutritional regulation by ants emerges from a distributed process: food is collected by a small fraction of workers, stored within the crops of individuals, and spread via local ant-to-ant interactions. The precise individual-level underpinnings of this collective regulation have remained unclear mainly due to difficulties in measuring food within ants' crops. Here we image fluorescent liquid food in individually tagged *Camponotus sanctus* ants and track the real-time food flow from foragers to their gradually satiating colonies. We show how the feedback between colony satiation level and food inflow is mediated by individual crop loads; specifically, the crop loads of recipient ants control food flow rates, while those of foragers regulate the frequency of foraging-trips. Interestingly, these effects do not rise from pure physical limitations of crop capacity. Our findings suggest that the emergence of food intake regulation does not require individual foragers to assess the global state of the colony.

DOI: https://doi.org/10.7554/eLife.31730.001

*For correspondence:
ofer.feinerman@weizmann.ac.il

[†]These authors contributed equally to this work

Competing interests: The authors declare that no competing interests exist.

## Introduction

Eusocial insects stand out in their ability to achieve collective regulation with no central control. Nutritional management in bees and ants is a compelling example. On the one hand, the colony as a whole displays high levels of collective regulation on the amount of food collected (*Howard and Tschinkel, 1980*; *Sorensen et al., 1985*; *Cassill and Tschinkel, 1999*), on its nutritional composition (*Dussutour and Simpson, 2009*; *Cook et al., 2010*; *Bazazi et al., 2016*), and on its dissemination within the colony (*Anderson and Ratnieks, 1999*; *Sendova-Franks et al., 2010*; *Greenwald et al., 2015*). On the other hand, this regulation is achieved by individuals that react to their local environment. Food dissemination often relies on local *trophallactic* interactions in which liquid food, not fully digested, is regurgitated from the crop of one individual and passed mouth-to-mouth to another ((*Hölldobler and Wilson, 1990*) chapter 7, page 291 and SI of (*Greenwald et al., 2015*)). Such distributed processes are characterized by intricate interaction networks that include significant random aspects (*Fewell, 2003*; *Pinter-Wollman et al., 2011*; *Mersch et al., 2013*; *Sendova-Franks et al., 2010*) which may hinder global coordination. How colonies manage to achieve tight nutritional regulation despite the difficulties that are inherent to a distributed process is not fully understood.

An essential component in the nutritional regulation of any living system is the adjustment of incoming food rates to the current level of satiation (*Parks, 2012*; *Simpson and Raubenheimer, 1993*; *Josens and Roces, 2000*). To experimentally approach the principles behind such adjustments it is useful to observe the global process in which food accumulates in the system. Indeed, the dynamics of food accumulation in ant colonies have been a subject of interest for many years (*Wilson and Eisner, 1957*; *Markin, 1970*; *Howard and Tschinkel, 1980*; *Buffin et al., 2009*; *Buffin et al., 2012*; *Sendova-Franks et al., 2010*; *Greenwald et al., 2015*). These studies show that when introduced to a new food source, the levels of food stored within the colony display logistic dynamics. The logistic growth in the amount of accumulated food supports the notion that total

**eLife digest** In an ant society, a small group of workers, called foragers, feeds the rest of the colony. Each forager goes out of the nest to find food; any liquid food she collects is stored in her 'crop', a pouch located just upstream of her stomach. When a forager goes back to the nest, she unloads this liquid by mouth-to-mouth contact into the crops of other ants.

The foragers need to adjust how often they go on foraging trips based on the amount of food the other ants require at any given time. However, it is still unclear how foragers can assess the changing needs of the colony. For example, it had been assumed that a forager would fully feed the individuals she encounters in the nest and then go for another foraging trip when her crop is empty. Yet, scientists had not managed to track food transfer at the level of the individual insect to confirm if this is the case.

Greenwald, Baltiansky and Feinerman have now used laboratory ant colonies and fluorescently labeled food to monitor in real time how food is transferred between individual ants. Contrary to previous hypotheses, when a forager comes back to the nest, she gives small portions of food to many different ants. The insects in the colony are therefore being nourished through these repetitive interactions. As the experiments show, when a forager meets other ants in the nest, the fullness of their crops reliably represents how full the colony is as a whole. Moreover, the portion that the forager gives is, on average, proportional to the space available in the receiver's crop: the emptier the crop, the more food is given. The amount of food in the crops of the receiving ants therefore controls how much food enters the colony, and the rate at which a forager unloads its crop.

A possible mechanism for regulating foraging frequency is that a forager considers whether or not to go on a foraging trip only after she senses a substantial change in the amount of food in her crop. In this case, her decision is based on the fullness of her own crop: the smaller the amount of food left in her pouch, the more likely she is to decide to leave the nest to bring in more food. Because the rate at which the foragers' crop empties is tied to the amount of food in the receiving ants' crops, how often the forager goes for food changes with the hunger level of the whole colony, with more trips when the ants are hungrier.

These experiments show that the amount of food in the crops of the receiving and foraging ants helps foragers adapt their behavior to the colony's needs. This mechanism means the insects can achieve a common goal without explicitly knowing it. However, it remains to be explained how exactly the mechanical changes in the fullness of foragers' crop underpin this decision-making process.

DOI: https://doi.org/10.7554/eLife.31730.002

food inflow is regulated by the amount of food already stored within the colony. The local origins of this global regulation are still not fully understood.

To understand how global food flow regulation emerges from single ant behaviors one should consider the forager ants. These ants, which typically constitute only a small fraction of all workers, are the ones responsible for bringing food into the nest (*Oster and Wilson, 1978*; *Traniello, 1977*). Therefore, any change in the global inflow of food to the colony must be manifested in the rate at which foragers collect and deliver food. Accordingly, colonies can regulate the inflow of food by modulating foraging effort: for example, by varying the number of active foragers through recruitment (*Gordon, 2002*). Indeed, many studies on the regulation of foraging have focused on recruitment behavior and have shown that it correlates with the colony's nutritional state (*Traniello, 1977*; *Seeley, 1989*; *Tenczar et al., 2014*; *Cassill, 2003*). In this work, we explore a less studied aspect of food flow regulation, namely, changes in the behavior of already active foragers. Active foragers engage in repeated trips between the food source and the nest (*Traniello, 1977*; *Tenczar et al., 2014*), where they use trophallaxis to deliver their food load to multiple recipients (*Seeley, 1989*; *Gregson et al., 2003*; *Huang and Seeley, 2003*; *Traniello, 1977*). The rate at which a forager leaves the nest for her next trip as well as the amount of food that she manages to unload per trip provide potential regulators of the collective foraging effort. These regulators may be tied to the colony's nutritional state through the experience of returning foragers when they unload in the nest. In this vein, it was shown that honeybee foragers experience longer waiting times between subsequent

unloading interactions if the colony is satiated (*Seeley, 1989*), and it was suggested that they use this information to adjust their recruitment behavior.

Most previous studies of individual forager behavior did not make direct connections between single ant rules and the global dynamics of food accumulation (*Seeley, 1989*; *Huang and Seeley, 2003*; *Gregson et al., 2003*). Nonetheless, the observations and interpretations they present are consistent with a simple intuition for the origins of the observed logistic dynamics in the accumulation of food: Initially, when a scout from a hungry colony encounters food she commences a recruitment process in which the number of active foragers increases (*Greene and Gordon, 2007*). This positive feedback is followed by a delayed negative feedback that results from the increased difficulty of foragers to locate available recipients as the colony satiates (*Seeley, 1989*; *Seeley and Tovey, 1994*; *Buffin et al., 2009*; *Sendova-Franks et al., 2010*). A simple prediction follows: if foragers wait to unload their entire crop contents before leaving for their next foraging trip (*Gregson et al., 2003*; *Traniello, 1977*) then the frequency at which a forager exits the nest should gradually decrease as the colony satiates (*Buffin et al., 2009*).

Although the above intuition may seem complete, it has only little empirical support. Until recently, microscopic measurements of real-time individual crop loads and food-flows in single interactions were unavailable. As a result, existing explanations for different aspects of the foraging process rely (either explicitly or implicitly) on various assumptions. The foragers were assumed to unload their entire crop contents before leaving the nest (*Traniello, 1977*; *Gregson et al., 2003*; *Buffin et al., 2009*) and use local experience to assess the colony's nutritional state (*Seeley, 1989*; *Seeley and Tovey, 1994*; *Huang and Seeley, 2003*). The recipients were assumed to be either empty or full (*Sendova-Franks et al., 2010*; *Seeley, 1989*; *Seeley and Tovey, 1994*) and fill upon a single interaction with a forager (*Sendova-Franks et al., 2010*; *Seeley, 1989*; *Seeley and Tovey, 1994*). As for the pattern of interactions between foragers and their recipients, it was assumed that in the nest a forager has a constant probability per unit time to interact with potential recipients (*Sendova-Franks et al., 2010*; *Seeley, 1989*; *Seeley and Tovey, 1994*), that there is a formation of queues of returning foragers and available receivers (*Seeley, 1989*), and that interaction patterns are random (*Seeley and Tovey, 1994*; *Buffin et al., 2009*; *Sendova-Franks et al., 2010*).

Relying on individual-level assumptions may be deceiving since multiple sets of microscopic rules can lead to similar macroscopic outcomes. For example, the slowing down of foragers' unloading rates may stem from reduced rates of trophallactic interactions but can also be the result of smaller amounts of food transferred per interaction. Both will affect the global outcome similarly. To uniquely identify the micro-scale mechanisms of food inflow regulation and examine the assumptions outlined above, we tracked fluorescently-labeled food in crops of individually tagged ants (*Greenwald et al., 2015*). This technology allowed for a non-intrusive study of the dynamics of food accumulation in ant colonies with a spatial resolution of single-ant crop loads and a temporal resolution sufficient to capture single trophallactic events. We thus present the missing experimental data on the crop contents of encountered ants, the amount of food transferred per interaction, the dynamics of forager unloading at different satiety states of the colony, and the amount of food in the foragers' crops when they exit the nest.

In the following sections we use these highly resolved measurements to quantitatively link the microscopic and macroscopic scales of food accumulation dynamics in ant colonies. We demonstrate how the global dynamics and the regulation of individual foraging effort rely on individual crop loads. Specifically, we delineate how individual crop loads affect a forager's unloading rate as well as her decision to exit the nest for the next food collection trip. Our findings suggest a distributed regulation mechanism which does not require individual foragers to assess global, colony-scale variables.

## Results

### Dynamics of food accumulation

Food accumulation dynamics were studied by introducing starved colonies of *Camponotus sanctus* ants to fluorescently-labeled food. Food was supplied *ad-libitum* to isolate the effects of colony satiation from the effect of resource availability on the inflow of food. As the colony replenished, we

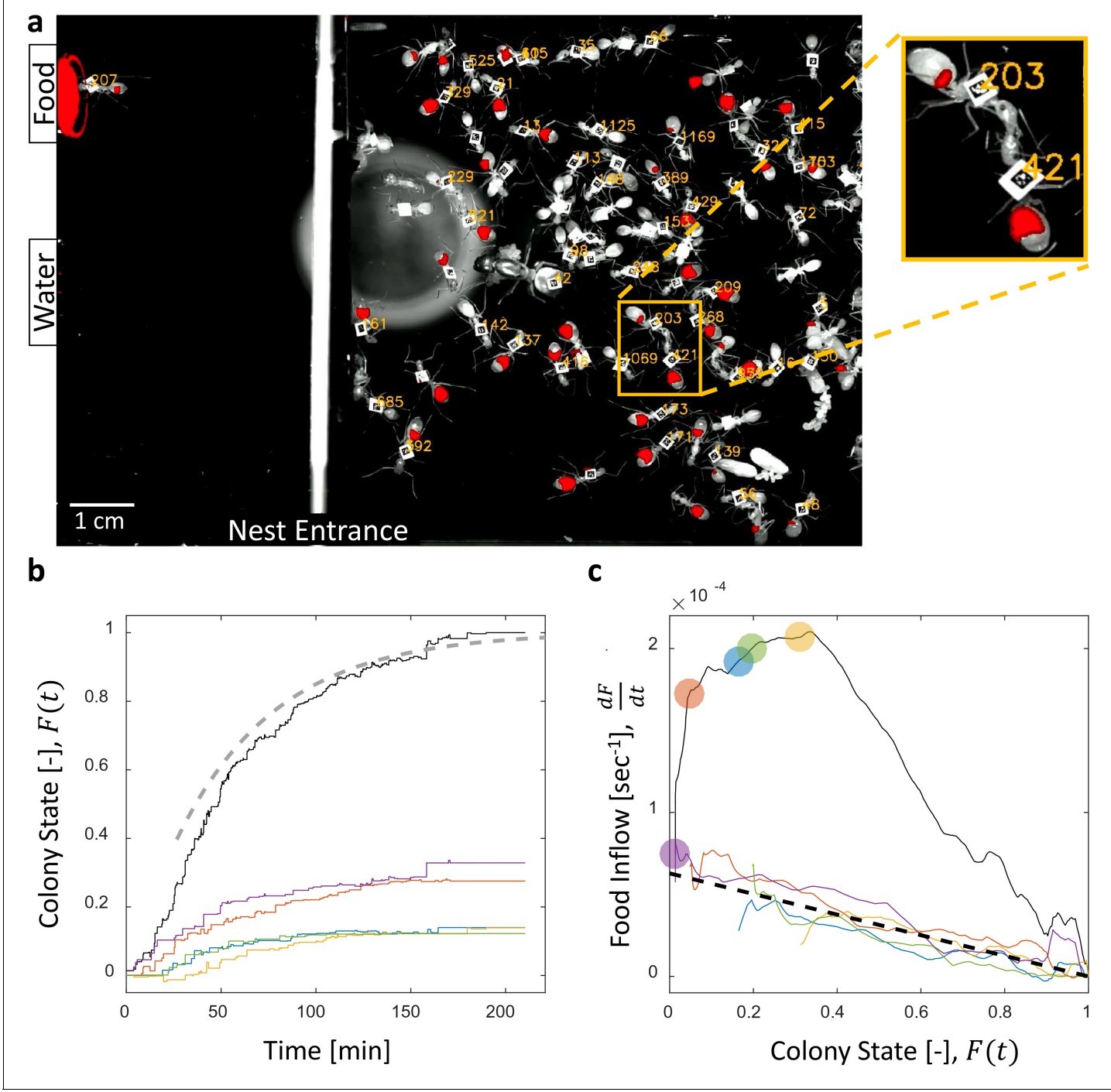

**Figure 1.** Dynamics of food accumulation in a starved colony. (**a**) A single frame from a video of a colony in the course of food accumulation. Ant identity is presented as a unique number next to her tag, and the fluorescent food is presented in red. The right side of the image is an IR-covered nest, and the left side is a neighboring open yard that includes a food source and a water source. In this frame a forager can be seen feeding from the food source (ant 207), and a trophallactic event between ants 203 and 421 is magnified. (**b**) Global food accumulation (normalized fluorescence), $F(t)$, is plotted in black. The accumulated food brought by each forager, $f_i(t)$ is plotted in a unique color. The dashed line is the predicted colony state according to **Equation 3** for $m = 0.6 \times 10^{-4}$, as estimated from **Figure 1c**. (**c**) The time-averaged global inflow, $\frac{dF}{dt}$, as a function of $F(t)$, is plotted in the black solid line. Time-averaged flows through individual foragers, $\frac{df_i}{dt}$, are plotted in unique colors (same as in panel b). Flows were calculated by differentiating the colony state and the contributions of each forager (the curves from **Figure 1b**) with respect to time (see Methods, Data Analysis). Colored circles on the global plot depict each forager's first return from the food source. The black dashed line represents **Equation 2**, where $m$ was calculated as follows: the flow through each forager was fit with an equation of the form $\frac{df_i}{dt} = m_i \cdot (1 - F(t))$, and $m$ was taken to be the average of all $m_i$.

*Figure 1 continued on next page*

*Figure 1 continued*

Results from all three experimental colonies can be found in *Figure 1—figure supplements 1* and *2*. Source files for panels b and c are available in *Figure 1—source datas 1* and *2*.

DOI: https://doi.org/10.7554/eLife.31730.003

The following source data and figure supplements are available for figure 1:

**Source data 1.** Trophallactic interactions.

DOI: https://doi.org/10.7554/eLife.31730.006

**Source data 2.** Temporal data.

DOI: https://doi.org/10.7554/eLife.31730.007

**Figure supplement 1.** Food accumulation dynamics in all three experimental colonies.

DOI: https://doi.org/10.7554/eLife.31730.004

**Figure supplement 2.** Food flows through individual foragers and their average.

DOI: https://doi.org/10.7554/eLife.31730.005

followed the traffic and storage of the food within the crops of individual ants using real-time fluorescent imaging (*Figure 1a*, *Video 1* and *Video 2*, for details see Materials and methods).

We found that the total amount of food in the colony gradually accumulated until saturation (*Figure 1b*, (*Buffin et al., 2009*)). The level at which food saturated was defined post-hoc as the colony's intake volume target. We define the 'colony state' at time $t$, denoted $F(t)$, as the total amount of food in the colony at time $t$ divided by the colony's intake target. The 'colony state' is thus a normalized measure of the colony's satiety level, starting from $F = 0$ when the colony is starved and gradually approaching $F = 1$ as the colony approaches its target (*Figure 1b*, black line).

To enable tracing of the food flow process on the single-ant level, all ants were individually tagged (*Figure 1a*, and see Methods, Experimental setup). As could be expected (*Gordon, 1989*; *Tenczar et al., 2014*), this labeling showed that a few consistent foragers were accountable for the transfer of food from the source to the ants in the nest (*Figure 1b*). This allowed us to study the dynamics of food accumulation by expressing colony state as the sum of the contributions of individual foragers:

$$F(t) = \sum_{i=1}^{N} f_i(t)$$

(1)

where $f_i(t)$ is the portion of the colony state contributed by forager $i$ by time $t$, and $N$ is the number of foragers.

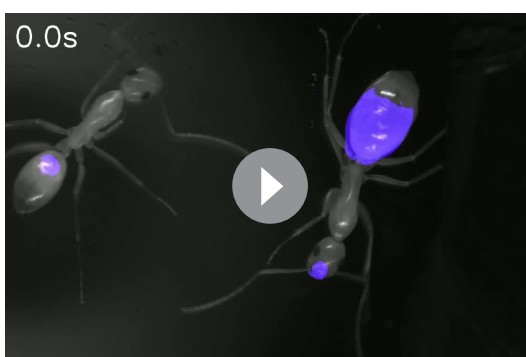

**Video 1.** A trophallactic event. Two *Camponotus sanctus* ants engaged in trophallaxis of fluorescent liquid food (presented in purple).

DOI: https://doi.org/10.7554/eLife.31730.008

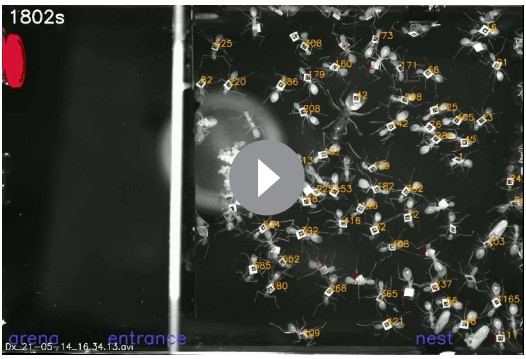

**Video 2.** Food accumulation within a colony of *Camponotus sanctus*. A starved colony replenishes on fluorescent liquid food (presented in red), brought in by few consistent foragers. Ant Identity is presented as a unique number next to her barcode tag.

DOI: https://doi.org/10.7554/eLife.31730.009

## Feedback on the individual forager scale

Collective food inflow ($\frac{dF}{dt}$), as well as the flow of food through each individual forager ($\frac{df_i}{dt}$) were derived by differentiating the measured colony state and the individual contributions with respect to time (*Figure 1c*, *Figure 1—figure supplement 2*, and Materials and methods, Data Analysis). This revealed that flows of food through each individual forager declined with increasing colony satiation state, $F$, and were roughly proportional to the available space in the colony, $1 - F$:

$$\forall i, \frac{df_i}{dt} \approx m(1 - F(t)) \tag{2}$$

where $m$ is a constant (*Figures 1c* and *Figure 1—figure supplement 2*). This linear relationship holds for each forager, regardless of when she began foraging and is thus incompatible with feed-forward control in which a forager slows down as a function of her own history. Rather, it supports a mechanism by which the colony state feeds back on the food transfer rate of each individual forager.

Breaking down the total inflow of food into individual forager contributions and using *Equation 2*, we obtain:

$$\frac{dF}{dt} = \sum_{i=1}^{n(t)} \frac{df_i}{dt} \approx n(t) \cdot m(1 - F(t)) \tag{3}$$

where $n(t)$ is the number of foragers that have begun foraging by time $t$. This formulation provides simple intuition for the non-monotonicity of food flow as apparent in *Figure 1c*. Specifically, an initial rise in collective inflow occurred when the number of foragers grew at a rate that overcame the rate of individual flow decay. Once the number of active foragers stabilized total flow rates declined linearly with colony state.

*Equation 3* describes a feedback process in which the rate of change in the colony state depends on the colony state itself, and more specifically - on $1 - F$, the space left to fill until the colony reaches its target. This is a direct consequence of individual foragers that deliver food at slower rates as the colony fills (*Equation 2*). However, the satiation state of the colony is a global factor that is, most likely, not directly available to individual ants. In the next section, we demonstrate how the observed feedback emerges from pairwise trophallactic interactions.

## Global feedback from local interactions

The average food flow through a single forager ($\frac{df_i}{dt}$) can be estimated by the product of two macroscopic parameters that may depend on the colony state, $F$: her average interaction rate, $\langle r(F) \rangle$, and the average volume transferred per interaction, $\langle v(F) \rangle$.

$$\frac{df_i}{dt} \approx \langle r(F) \rangle \cdot \langle v(F) \rangle \tag{4}$$

While both the interaction rate and the interaction volume declined with increasing colony state, the change in the interaction volume was more prominent (*Figure 2a*, *Figure 2—figure supplements 1* and *2*). In fact, interaction volumes were nearly sufficient to account for the inflow dynamics, while the interaction rate introduced a minor second-order correction (*Figure 2b* and *Figure 2—figure supplement 3*). Importantly, interaction rates alone did not suffice to account for the inflow dynamics (*Figure 2b*).

Therefore, we turned to explore the local determinants that affect interaction volumes, under the assumption that interaction volumes are set locally depending on the states of the interacting individuals. The maximal potential volume of any given interaction is constrained by both the donor's crop load and the available space in the recipient's crop. To inspect the impact of each of these two local factors, we examined the distribution of all interaction volumes ($v$) from foragers to non-forager recipients for different ranges of crop loads, either of the recipient ($c_{recipient}$, *Figure 3—figure supplement 1*) or of the forager ($c_{forager}$, *Figure 3—figure supplement 2*). We found that these distributions all follow an exponential probability density function (PDF) of the form:

$$p(v|c) = \lambda_c e^{-\lambda_c v} \tag{5}$$

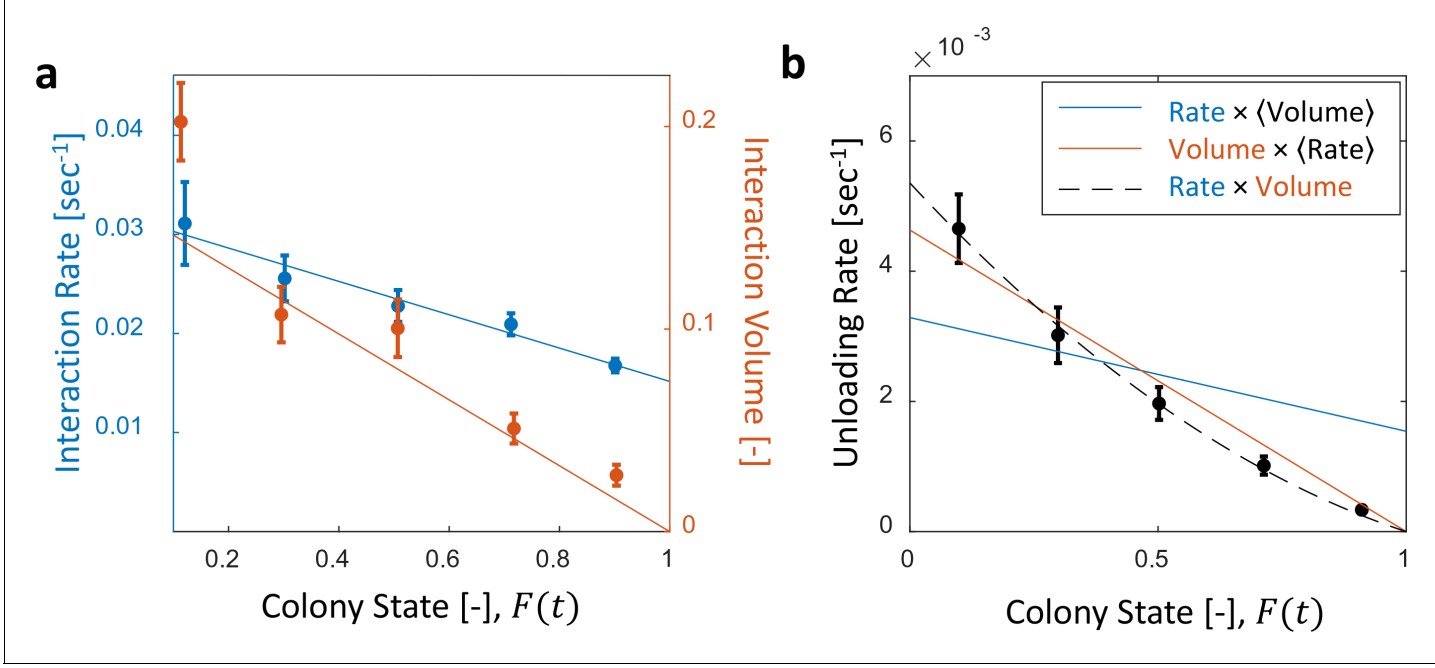

**Figure 2.** Interaction volume is the dominant component of unloading rate. (a) Interaction rate (blue) and interaction volumes (red) both decline with increasing colony state. Binned data is presented by mean ± SEM. Interaction rate was calculated as the inverse of all intervals between interactions. The intervals were binned according to the colony state at which they occurred (n = 49, 79, 101, 170, 240 for bins 1–5, respectively. See *Figure 2— figure supplement 1* for raw data and *Figure 2—source data 1* for binning sensitivity analysis). Blue line depicts a linear fit $r(F) = 0.032 - 0.017F$, $R^2 = 0.96$. Interaction volumes were measured in units of pixel intensities, normalized between experiments (see Methods, Data Analysis), and binned into equally-spaced colony state bins (n = 84, 137, 165, 274, 496 for bins 1–5, respectively. See S4). Red line represents the predicted relationship between the mean interaction volume and the colony state from *Equation 9*, with $C_0 = 1.14$ and $\frac{1}{\lambda_0} = 0.14$ as obtained from the fit in *Figure 3a*. (b) Foragers' unloading rates at each visit in the nest were binned according to colony state (black, mean ± SEM for each bin, n = 26,26,28,39,57 for bins 1–5, respectively). Mean unloading rate values were fitted by three functions: the blue line represents a model which includes the effect of interaction rates only (*unloading rate* $\propto 0.032 - 0.017F$, function obtained from fit in panel a, $R^2 = 0.52$), the red line represents a model which includes the effect of interaction volumes only (*unloading rate* $\propto 0.2 - 0.2F$, function obtained from fit in *Figure 2—figure supplement 2*, $R^2 = 0.96$), and the black dashed line represents a model that incorporates the combined effects of interaction volumes and interaction rates (*unloading rate* $\propto (0.032 - 0.071F) \cdot (0.2 - 0.2F)$, $R^2 = 0.99$). All panels in this figure represent pooled data from all three observation experiments. For raw data see *Figure 2—figure supplement 3*. Source file is available in *Figure 1—source data 1*.

DOI: https://doi.org/10.7554/eLife.31730.010

The following source data and figure supplements are available for figure 2:

**Source data 1.** Sensitivity analysis for binning interaction rate data.

DOI: https://doi.org/10.7554/eLife.31730.014

**Figure supplement 1.** Raw (gray) and binned (black) data of interaction rates.

DOI: https://doi.org/10.7554/eLife.31730.011

**Figure supplement 2.** Raw data pooled from all three observation experiments (gray) and binned (black) data of interaction volumes.

DOI: https://doi.org/10.7554/eLife.31730.012

**Figure supplement 3.** Raw (gray) and binned (black) data of unloading rates.

DOI: https://doi.org/10.7554/eLife.31730.013

where $p(v|c)$ is the conditional PDF of interaction volumes, $v$, given a crop load $c$, and $c$ is either $c_{forager}$ or $c_{recipient}$. We found that while the recipient's crop load affected the distribution of interaction volumes (*Figure 3a*), that of the forager had little effect, if any (*Figure 3b*). Specifically, the distribution of interaction volumes scaled with the space left to fill in the recipient's crop but was effectively independent of the forager's crop load (hence, hereafter we take $c = c_{recipient}$):

$$\lambda_c = \frac{\lambda_0}{C_0 - c} \tag{6}$$

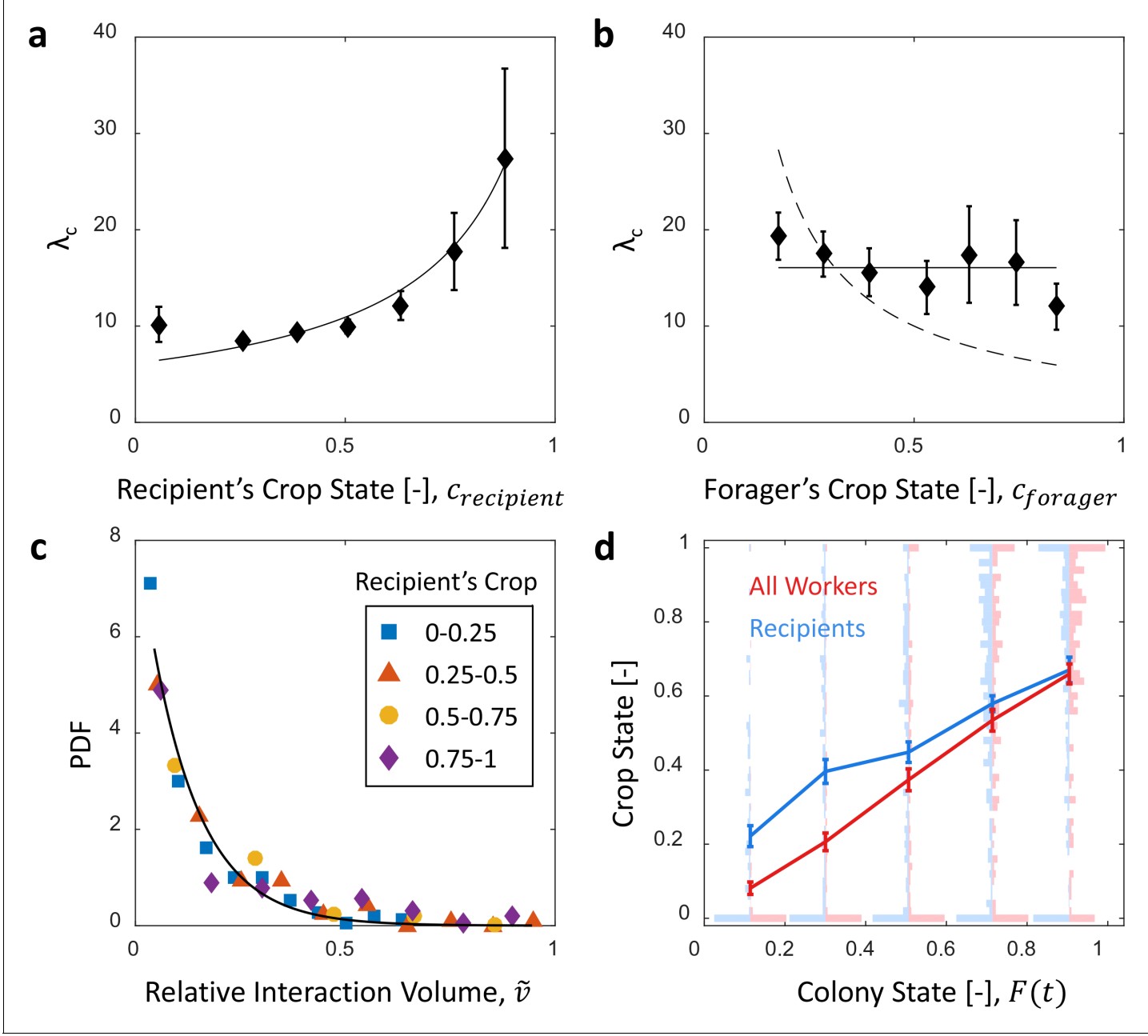

**Figure 3.** Microscopic food flow. (a) The distributions of interaction volumes from foragers to non-forager recipients, at seven ranges of *recipient* crop loads, follow an exponential probability density function of the form $p(v|c) = \lambda_c e^{-\lambda_c v}$ (*Equation 5*, *Figure 3—figure supplement 1*). Here the seven rate parameters of the distributions ($\lambda_c$) are plotted as a function of the *recipient's* crop load ($c_{recipient}$). Mean ± STD of $\lambda_c$ are over different binnings of the histograms to which the exponential distribution function was fit (17 histogram binwidths uniformly covering the range [0.01–0.09]). Curve represents a fit of the function $\lambda_c = \frac{\lambda_0}{C_0 - c}$ (*Equation 6*): $\lambda_0 = 7.01$, $C_0 = 1.14$, $R^2 = 0.93$. (b) Similarly to panel a, the rate parameters of the exponential distributions of interaction volumes were obtained for seven ranges of *forager* crop loads (*Figure 3—figure supplement 2*) and plotted as a function of the *forager's* crop load, $c_{forager}$. Dashed curve represents a fit of the form $\lambda_c = \frac{const}{c_{forager}}$, similar to *Equation 6*, but instead of a fraction of the recipient's empty crop space, $\lambda_c$ is assumed to be a fraction of the forager's crop load. $R^2$ was negative, indicating that this function is no better fit to the data than a constant (solid line). (c) The distributions of 'relative interaction volumes', $\tilde{v}$, wherein each interaction volume was normalized to the available space in the receiver's crop ($\tilde{v} = \frac{v}{C_0 - c}$). The distributions collapse onto a single exponential function $p(\tilde{v}) = \frac{1}{\lambda} e^{-\frac{1}{\lambda} \tilde{v}}$, $\lambda = 0.12$, $R^2 = 0.96$. (d) The crop loads of non-foragers at interactions with foragers (blue, n as in b) compared to crop loads of all non-foragers in the colony (red, n = 202 per colony state bin). The distributions of crop loads at each colony state are plotted as a violin plot, and the mean ± SEM are plotted in solid lines. All panels in this figure represent pooled data from all three observation experiments. Source file is available in the *Figure 1—source data 1*.

*Figure 3 continued on next page*

*Figure 3 continued*

DOI: https://doi.org/10.7554/eLife.31730.015

The following figure supplements are available for figure 3:

**Figure supplement 1.** Interaction volume distributions given recipient crop load.

DOI: https://doi.org/10.7554/eLife.31730.016

**Figure supplement 2.** Interaction volume distributions given Forager's crop load.

DOI: https://doi.org/10.7554/eLife.31730.017

**Figure supplement 3.** Mean interaction volumes as a function of the recipient's crop load.

DOI: https://doi.org/10.7554/eLife.31730.018

where $\lambda_0 = 7.01$ and $C_0 = 1.14$ (*Figure 3a*). $C_0$ may be interpreted as the average crop load target to which recipients aim to ultimately fill (expected to be close to 1).

On average, in an interaction with a forager the recipient receives $\frac{1}{\lambda_0} = 0.14$ of the space left to fill in her crop. Consistently, a linear fit relating mean interaction volume to recipient crop load $\langle v(c) \rangle = ac + b$ (*Figure 3—figure supplement 3*) yields $b \approx -a \approx 0.13 \approx \frac{1}{\lambda_0}$. This is to be expected if $\langle v(c) \rangle = \frac{1}{\lambda_0}(1 - c)$. Indeed, normalizing interaction volumes by the total amount of available space in the recipient's crop, $\tilde{v} = \frac{v}{C_0 - c}$, we find that all trophallactic volume distributions collapse onto a single exponential function $p(\tilde{v}) = \lambda e^{-\lambda \tilde{v}}$ with $\frac{1}{\lambda} = 0.12 \approx \frac{1}{\lambda_0}$ (*Figure 3c*). Simply put, the volume of an interaction can be estimated by a random exponentially distributed fraction ($\tilde{v}$) of the available space in the recipient's crop ($C_0 - c$).

We can now move forward to express mean interaction volume, $\langle v \rangle$, in terms of the colony state. Defining $p(v|F)$ to be the conditional probability for an interaction of volume $v$ when the colony state is $F$, the mean interaction volume (at colony state $F$) can be calculated by:

$$\langle v(F) \rangle = \int_v v \cdot p(v|F) dv \tag{7}$$

In light of our findings that interaction volumes change mainly with respect to the recipient's crop load, we can decompose $p(v|F) = \int_c p(v|c) \cdot p(c|F) dc$ where $p(c|F)$ is the probability density that the recipient will have a crop load of size $c$ at colony state $F$. *Equation 7* now becomes:

$$\langle v(F) \rangle = \int_c p(c|F) \int_v v \, p(v|c) \, dv \, dc \tag{8}$$

The probability ($p(c|F)$) changed as the colony satiated and individual ants approached their targets. *Figure 3d* shows that the ants that interact with a forager reliably represent the satiation level of the colony and that the accuracy of this representation increases as the colony satiates. Altogether, substituting the microscopic interaction rule described by *Equations 5 and 6* into the global summation described by *Equation 8* demonstrates how the average interaction volume changes in proportion to the empty space in the colony $(1 - F)$:

$$\langle v \rangle = \int_c p(c|F) \int_v \frac{\lambda_0}{C_0 - c} e^{-\frac{\lambda_0}{C_0 - c} v} v \, dv \, dc = \int_c p(c|F) \frac{C_0 - c}{\lambda_0} \, dc = \frac{1}{\lambda_0}(C_0 - \langle c(F) \rangle) = \frac{C_0}{\lambda_0}(1 - F) \tag{9}$$

where the following identities were used: $C_0 = \frac{\sum c_{target}}{N}$, $F = \frac{\sum c}{\sum c_{target}}$, $\int p(c) dc = 1$ and $\int p(c) c dc = \langle c \rangle$. For each ant, $c_{target}$ signifies her crop load at colony satiation. The value of the multiplicative factor $\frac{C_0}{\lambda_0} \approx 0.16$ stands in agreement with our experimental measurements (*Figure 2—figure supplement 2*).

The above analysis demonstrates how the global inflow is determined by interaction volumes that are locally controlled by the recipient's crop load, which on average represents the colony state. The forager's crop comes into play in a different aspect of food inflow. Its finite capacity requires foragers to repeatedly leave the nest to reload at the food source in order to supply food to the entire colony. However, leaving the nest encompasses inevitable risks and energetic costs. Therefore, it is

interesting to study whether foraging effort is also regulated, and if so, how it is expressed in decisions of individual foragers to leave the nest.

## Foraging effort is matched to the colony's Needs

To check whether foragers adjust their activity to the changing colony state we examined their behavior with respect to the accumulating food in the colony. The behavior of individual foragers was typically of a cyclic nature, as they alternated between two phases: (1) an outdoor phase, in which they filled their crops at the food source, and (2) an indoor phase, in which they distributed their crop contents in trophallaxis to colony members inside the nest (*Figure 4a*).

The frequency of these cycles ('foraging frequency') displayed a tight linear relationship with the available space in the colony $(1 - F)$, demonstrating that foraging effort is matched to the colony's needs (*Figure 4b* and *Figure 4—figure supplement 1a*, $y = 0.8 \ 10^{-3} + 3.6 \ 10^{-3}(1 - F)$, $R^2 = 0.98$). Additionally, the increase in cycle times was mainly attributed to the prolonged indoor phase of the cycle, rather than the relatively constant outdoor phase (*Figure 4c* and *Figure 4—figure supplement 1b*, Spearman's correlation test, indoor phase: $r_s = 0.77$, $p<0.001$, outdoor phase: $r_s = 0.13$, $p = 0.08$). This suggests that foraging frequency was regulated by the colony.

To test for a causal effect of colony state on foraging frequency, we elicited an external perturbation on the colony state. Indeed, in experiments where the colony state was actively dropped by introducing new hungry ants after others had reached satiation (see Methods, Perturbation Experiment), foragers' durations in the nest sharply dropped as well (*Figure 4e* and *Figure 4—figure supplement 1*). This response generated a secondary rise in the amount of food in the colony, relaxing at a new value as durations in the nest gradually lengthened once again (*Figure 4d–e* and *Figure 4—figure supplement 2*). These experiments explicitly decoupled colony state from the time that passed since the initial introduction of food, and thus show that the colony state rather than forager history was the important factor that affects foraging frequency.

Overall, these findings portray the following negative feedback process: foragers raise the colony state by bringing in food, while the colony state, in turn, inhibits their foraging frequency (*Figure 4f*). A possible mechanism for this feedback might be that foragers do not exit the nest for their next foraging trip before they fully unload (*Traniello, 1977*; *Gregson et al., 2003*; *Buffin et al., 2009*). In this case unloading rates directly dictate the foraging frequency. This is consistent with the fact that both unloading rate and the foraging frequency are proportional to the total available space, $1 - F$. We explore this hypothesized mechanism in the next section.

## Foragers' crop loads upon leaving the nest

We find that foragers do not leave the nest only after they have fully unloaded (*Figure 5a*). Nevertheless, the average amount of food in their crops when they exit remains nearly constant over different colony states (*Figure 5a*, Spearman's correlation test, $r_s = 0.24$, $p = 0.001$). This constant averagesuffices in producing the observed relation between foraging and unloading rates as specified above.

To maintain a relatively constant average crop state upon exit despite the declining unloading rates, foragers stayed longer in the nest (*Figure 4c*) and performed more interactions (*Figure 5c*, and *Figure 5—figure supplement 1a*). They also actively explored deeper into the nest (*Figure 5d* and *Figure 5—figure supplement 1b*). Surprisingly, even though the average crop load with which foragers exit the nest remains constant, this is not because the foragers unload a fixed amount in each visit. Rather, the crop loads with which foragers left the nest were highly variable (*Figure 5b*). This raises the questions of when foragers decide to exit and how these decisions lead to exits at variable crop contents that, nevertheless, maintain a constant average over different colony states. Do factors other than their own crop load affect their decisions to exit, and more specifically, do the foragers use high-level information regarding the colony state?

## Forager exit times are determined by their crop load and their unloading rate

To gain insight on the role of individual versus collective information in foragers' decisions to exit, we were interested in a forager's probability to exit the nest as a function of her own crop state (*crop*) and the colony state (*colony*). We first estimated the probability for an individual forager to exit

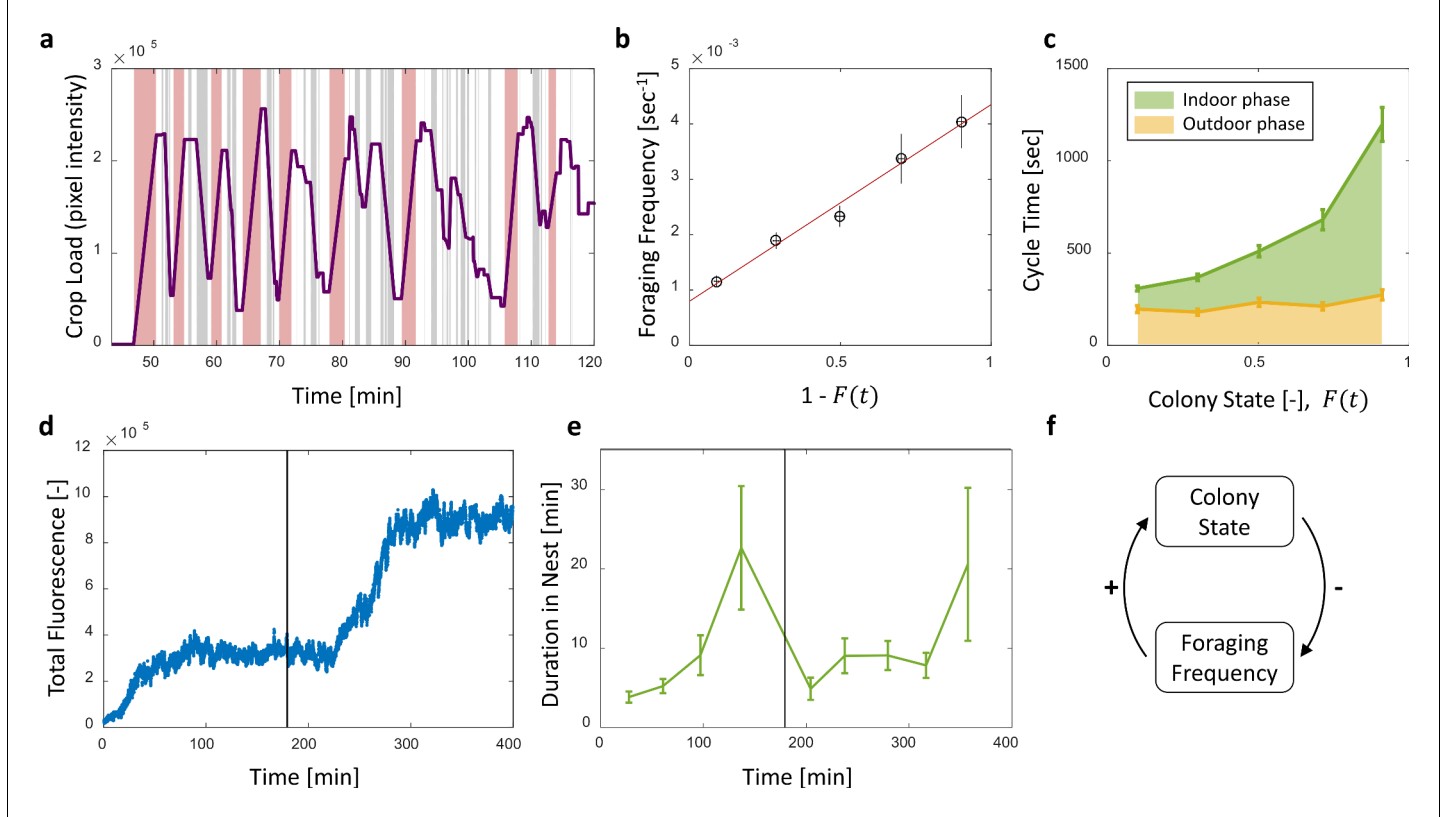

**Figure 4.** Foraging cycles. (**a**) The estimated crop load of a single forager during the first two hours of an experiment. As typical for a forager, her crop load oscillates as she alternates between feeding at the food source (pink areas) and unloading in trophallaxis (gray areas) in continuous back-and-forth trips. (**b**) The foraging frequency of individual foragers, calculated as the inverse of cycle times (the time interval between two consecutive feeding events of a single forager), grows linearly with the empty space in the colony, $1 - F$. Data points and error bars represent means and SEM of cycles. The pooled data from all three observation experiments is grouped into equally-spaced bins of colony state (n = 57,39,28,26,26, for bins 1–5, respectively, see **Figure 4—figure supplement 1**). A linear fit is presented in red: $y = 0.8 \ 10^{-3} + 3.6 \ 10^{-3}(1 - F)$, $R^2 = 0.98$. (**c**) Forager cycle durations are composed of an indoor phase (green) and an outer phase (yellow), the former accounting for most of the rising trend. The pooled data from all three observation experiments was binned and averaged as in panel b (n = 26,26,28,39,57, for bins 1–5, respectively, **Figure 4—figure supplement 1**). (**d**) Food accumulation in a perturbation experiment. Food rises to an initial plateau, and rises again to a secondary plateau after new hungry ants are introduced (black line). (**e**) Durations of foragers in the nest in the manipulation experiment described in panel d. Durations grow longer, drop after new hungry ants are introduced (black line), and subsequently rise again. Data points and error bars represent means and standard errors of durations of cycles grouped into time bins (n = 28,36,21,14,9,28,27,19,5, for bins 1–9, respectively). Raw data and results from a second replication of the perturbation experiment are presented in **Figure 4—figure supplement 2**. (**f**) A schematic representation of the observed negative feedback between the colony state and the foraging frequency. Source file for panels b and c is available in the **Figure 4—source data 1**. Source file for panels d and e is available in the **Figure 4—source data 2**.

DOI: https://doi.org/10.7554/eLife.31730.019

The following source data and figure supplements are available for figure 4:

**Source data 1.** Foraging cycles.
DOI: https://doi.org/10.7554/eLife.31730.022
**Source data 2.** Manipulation experiments.
DOI: https://doi.org/10.7554/eLife.31730.023
**Figure supplement 1.** Foraging cycle times.
DOI: https://doi.org/10.7554/eLife.31730.020
**Figure supplement 2.** Perturbation experiments.
DOI: https://doi.org/10.7554/eLife.31730.021

per time unit given her crop load and colony state $R(exit \mid crop, colony)$ (hereinafter, $R_{exit}$). We found that this exit rate was strongly dependent on both the forager's crop state and the colony state (**Figure 6a**, **Table 1**).

To understand which information the foragers require to generate this exit pattern we make the simplifying and common assumption that $R_{exit}$ is an outcome of a Markovian decision process (**Robinson et al., 2011**; **Sumpter et al., 2012**), and can be treated as a product of two probabilities:

$$R(exit \mid crop, colony) = R(make\ a\ decision) \cdot P(decision = exit \mid crop, colony) \qquad (10)$$

where $R(make\ a\ decision)$ is the probability of a forager to make a decision within a time unit (hereinafter, $R_{decide}$), and $P(decision = exit \mid crop, colony)$ is her probability to decide to exit given her crop load and colony state when a decision is made (hereinafter, $P$).

Since the precise timings of an ant's decisions are beyond our experimental reach, we replaced $R_{decide}$ by three assumed decision rates: (1) a constant decision rate, (2) a decision rate that is matched to the forager's interaction rate, and (3) a decision rate matched to the forager's unloading rate. **Figure 6** shows the corresponding $P$ for each decision rate. For a constant decision rate, $P$ is proportional to $R_{exit}$ and depends on both the forager's crop state and the colony state (**Figure 6a**, **Table 1**). For a decision rate that is matched to the forager's interaction rate (e.g. a forager considers whether to exit only after an interaction has ended), $P$ was calculated by considering only observations at ends of interactions as decision points. In this case, the effect of the colony state on $P$ is present but smaller that the effect of the forager's crop (**Figure 6b**, **Table 1**). Last, for a decision rate matched to the forager's unloading rate, $P$ was calculated by considering observations at fixed intervals of the forager's crop load as decision points. For this case, the effect of the colony state on $P$ approaches zero, such that $P$ varies predominantly with the forager's internal crop state (**Figure 6c–d**, **Table 1**).

Since the probability to decide to exit, $P$, was effectively independent of the colony state when the rate of decisions, $R_{decide}$, was adjusted to the unloading rate, we learn that the rate of exits $R_{exit}$ can be decomposed into two functions with a clear separation of variables:

$$R(exit \mid crop, colony) = U(colony) \cdot G(crop) \qquad (11)$$

where $U(colony)$ is linear in the unloading rate (**Equation 2**) and $G(crop)$ is a function of the crop that does not depend on the colony state (**Figure 6d**). Interpreting this result from the perspective of the individual forager suggests a simple biological mechanism that may underly this separation of variables: In the course of unloading, the forager considers whether to exit or not each time she senses a sufficiently large change in her crop load, and then decides to exit based on her crop load alone. Since the rate at which her crop load changes is mainly affected by the recipients (**Figures 2** and **3**), the rate of her decisions is controlled by the colony ($U(colony) \propto 1 - F$); once the forager is triggered to make a decision, the decision itself depends on personal information alone ($G(crop)$, **Figure 6d**).

## Discussion

The total flow of food into an ant colony is the sum of many small trophallactic events between foragers and recipient ants. A reliable description of colony level food regulation therefore demands empirical data on the statistics and rules that guide these microscopic events. In this work, we provide the first quantitative multi-scale account of the dynamics of ant colony satiation that links the global state to local interaction statistics. While our global-scale measurements generally concur with previous studies, our microscopic observations revealed two significant deviations from prevalent individual-level assumptions. First, foragers do not necessarily deliver their entire crop load before exiting the nest. Second, recipients do not fill to their capacity in a single interaction. The next subsections discuss the ways in which our new individual-level observations agree with previous collective level measurements despite the aforementioned deviations. We further discuss how these new findings alter the current understanding of the food intake process.

### Regulation of food flow

On the scale of the entire colony, and in agreement with previous studies (**Buffin et al., 2009**; **Sendova-Franks et al., 2010**), we find that food accumulation follows logistic dynamics (**Figure 1b**). Our individual-level measurements confirm that the logistic equation which describes the global dynamics can be interpreted as the product of two intuitive terms (**Equation 3**): the number of active

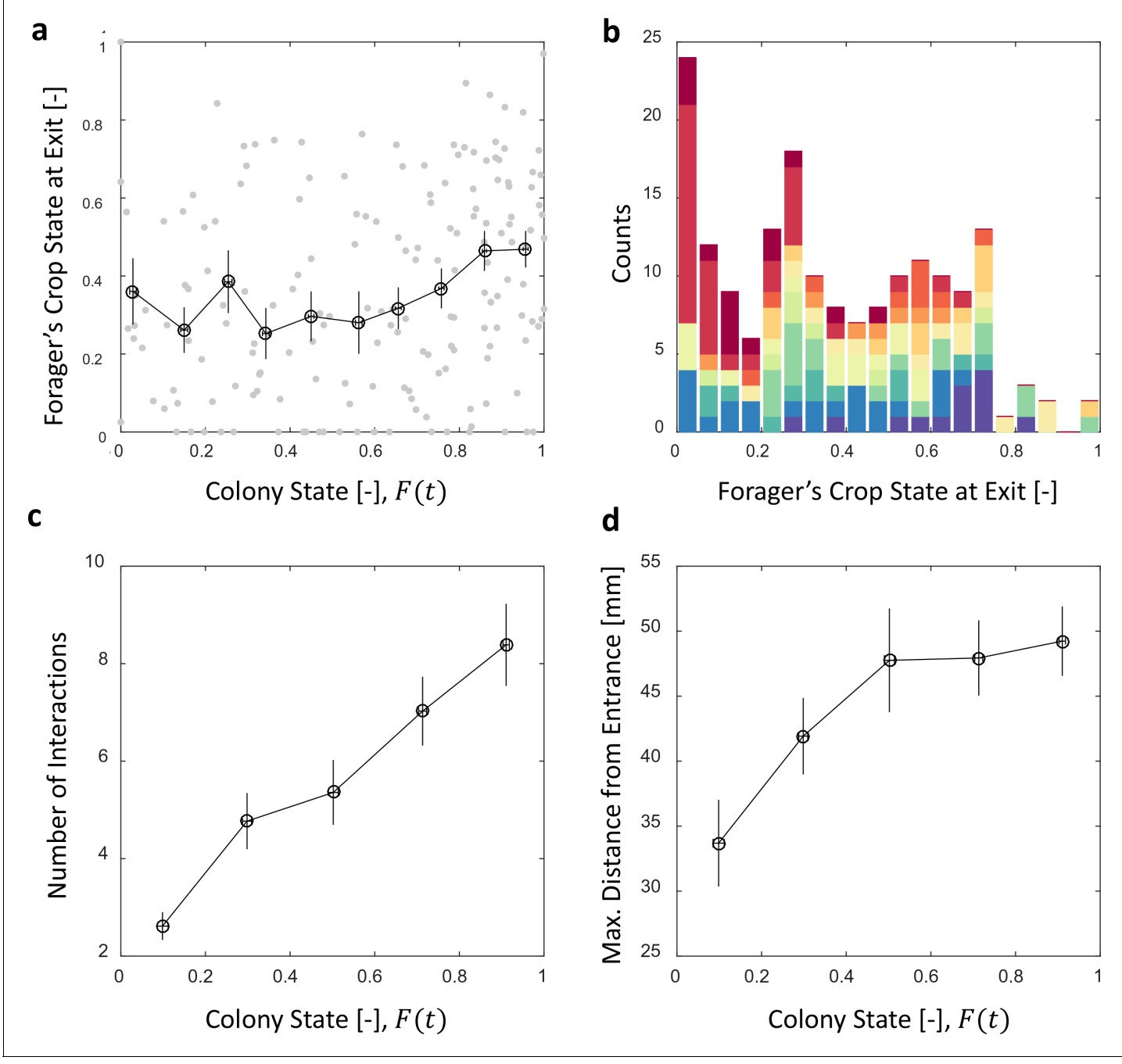

**Figure 5.** Foragers' crop loads at exit. (a) Foragers' crop loads at the moments of exit were only weakly dependent on colony state (Spearman's correlation test, $r_s = 0.24$, $p = 0.001$), the average remaining approximately constant with only a slight rise at high colony states. Gray: raw data, black: mean ± SEM of binned data (n = 11,15,13,13,15,13,17,22,27,30 for bins 1–10, respectively). (b) The wide distribution of foragers' crop loads at the moments of exit (n = 176). Each color represents a different forager, revealing that the distribution of crop loads upon exit is wide within each forager and not due to inter-individual variability. (c) The number of interactions a forager has in a single visit to the nest rises as the colony satiates, mean ± SEM of binned data (n = 26,26,28,39,57 for bins 1–5, respectively, *Figure 5—figure supplement 1a*). (d) Foragers reach deeper locations in the nest as the colony satiates, mean ± SEM of binned data (n as in panel c, *Figure 5—figure supplement 1b*). All panels relate to the pooled data from all three observation experiments. Source file is available in the *Figure 4—source data 1*.

DOI: https://doi.org/10.7554/eLife.31730.024

The following figure supplement is available for figure 5:

**Figure supplement 1.** Number of interactions and maximal distance from entrance in a forager's visit in the nest.

DOI: https://doi.org/10.7554/eLife.31730.025

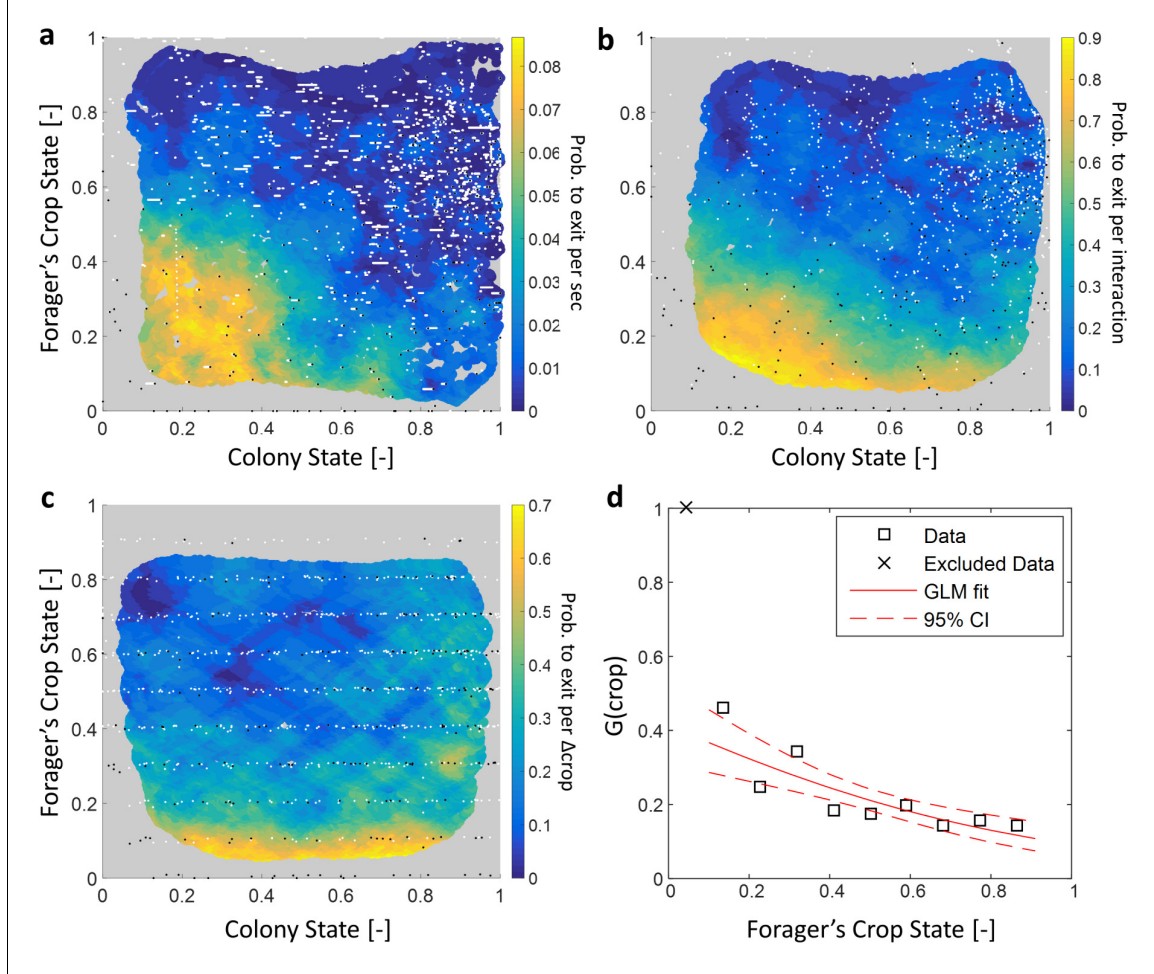

**Figure 6.** Forager exits. (a–c) Forager exit probabilities as a function of the colony state and her own crop state. All panels relate to the pooled data from all three observation experiments. Observations are plotted on a 2-dimensional space of the forager's crop state and the colony state, as black and white dots (white - 'stay' observations, black - 'exits'). An observation was classified as an 'exit' if the forager left the nest before the next considered observation. The colored surface represents the estimated local probability to exit on this space, calculated as the fraction of 'exits' out of all observations in each bin in the space: the color of each pixel on the surface represents the probability calculated based on the $n$ closest data points, and the pixel's location is the average location of these points (a: $n = 300$, b-c: $n = 30$). The three panels consider three possible decision rates: Constant decision rate (a) where all observations, taken every two seconds, were considered to be decision points (excluding observations during trophallaxis). Decision rate matched to interaction rate (b) where only observations at ends of interactions were considered to be decision points. Decision rate matched to unloading rate (c) where only observations taken each time a forager unloaded $\Delta crop$ food, were considered to be decision points ($\Delta crop$=10% of the forager's capacity). (d) $G(crop)$, a projection of the two-dimensional probability presented in panel c on the forager's crop state axis. Since foragers' crop loads rarely rose in the nest, their lowest crop observation in a visit was generally an 'exit', so the calculated probability to exit in the lowest crop interval was 1. To ensure that the crop state played a role beyond this extreme effect, the GLM fit did not include the lowest crop interval. The crop load effect was significant: $\beta_c = -1.93$, $p<0.001$. Source file for panels a and c is available in the *Figure 1—source data 2*. Source file for panel b is available in the *Figure 1—source data 1*.

DOI: https://doi.org/10.7554/eLife.31730.026

foragers times the average unloading rate per forager. As previously speculated (*Buffin et al., 2009*), the initial rise in the global food inflow stems from gradually joining foragers while its subsequent decay is the result of a negative feedback process wherein colony satiation levels work to decrease the unloading rates of individual foragers (*Figure 1c*).

We traced the mechanisms of this large-scale negative feedback to the immediate experience of individual foragers and specifically to the crop-loads of the ants they interact with. When a forager enters the nest she interacts with a 'representative sample' of recipient ants, i.e. ants whose crop load is, on average, proportional to the total satiation state of the colony (*Figure 3d*). Further, in

each such interaction the amount of food transferred is random but, on average, proportional to the available space in the recipient's crop (*Figure 3a,c*). Together, these findings imply that unloading rates are determined by the colony and directly proportional to the total empty space in the crops of the entire colony.

Many sets of local rules could have yielded the same average flow, and thus would have been consistent with similar global dynamics. For example, previous studies have attributed the global negative feedback to the decreasing probability of a forager to encounter an accepting recipient, which delays the time until delivery (*Sendova-Franks et al., 2010*; *Seeley, 1989*; *Seeley and Tovey, 1994*; *Cassill and Tschinkel, 1999*). However, due to experimental limitations these studies relied on the implicit assumption that recipients are satisfied by a single interaction, while in fact recipients may very well be partially satiated (*Huang and Seeley, 2003*). Our measurements on the level of single crops show that recipients are typically partially loaded, and the effect of their crop loads on interaction sizes is more dominant than the minor decrease of interaction rates in generating the collective negative feedback.

Interestingly, partial crop loads do not affect the interaction volume merely by physical limitation: in most interactions the donor does not deliver her entire crop load, nor does the recipient fill up to her capacity. This finding contradicts the prevalent assumption used by those studies that did take partially loaded recipients into account (*Huang and Seeley, 2003*; *Gregson et al., 2003*). These studies supposed that the amount of food transferred in an interaction is the maximal possible amount, and partial crop loads result from discrepancies between foragers' loads and recipients' capacities. Here we introduce explicit measurements of interaction volumes that reveal that exponentially distributed interaction volumes lead to partially loaded ants. This volume distribution concurs with the global feedback as it is scaled to the available space in the recipient's crop. Feedback based on interaction volumes that are not set by physical limitations, rather than 'all-or-none' interactions, potentially allows individual ants to fine-tune their intake and allow for combinations of several sources towards their desired nutritional target (*Cassill and Tschinkel, 1999*).

## Regulation of foraging trips

Previous studies typically addressed the global feedback between colony state and the collective foraging effort (*Seeley, 1989*; *Seeley and Tovey, 1994*; *Cassill, 2003*). Our work complements this by demonstrating how this feedback acts on individual foraging frequencies (*Figure 4*, see also [*Tenczar et al., 2014*] and [*Rivera et al., 2016*]). Furthermore, while previous studies suggested that foragers use local information, such as time delays, interaction rates or number of refused interactions, to infer the colony's needs (*Seeley, 1989*; *Seeley and Tovey, 1994*; *Cassill, 2003*; *Greene and Gordon, 2007*; *Gordon et al., 1993*), we propose a mechanism that demonstrates how foragers could adjust their foraging frequency relying on their own crop load alone (*Figure 6c,d*) (*Mayack and Naug, 2013*). In brief, foragers could adjust their exit rates to colony needs by modulating their decision rate according to unloading rates, while the decision itself depends on their current crop load alone.

Interestingly, foragers usually do not exit completely empty (*Figure 5a,b*) as could be intuitively assumed (*Gregson et al., 2003*; *Buffin et al., 2009*). This provides further evidence that, similar to interaction volumes, foraging activity is not regulated by pure physical limitations (i.e. an empty crop). Rather, we have found that foragers exit with a wide range of crop loads. The lack of a well-defined exit threshold entails a potentially wasteful effect in which forager crop loads at exit increase with colony state: The difficulty of unloading at higher colony states means that foragers spend longer times with a relatively full crop. Since there is a probability to exit at any crop load this may lead to an upward drift in the crop loads of exiting foragers. Here we show this drift is minor (*Figure 5a*) and propose different options by which this may be achieved. For example, it could be the case that after each interaction a forager decides whether to exit the nest or, rather, wait for another opportunity to unload. This decision scheme holds an appealing simplicity as it implies that a forager's decision rate is set externally and not by an internal clock or parameters. On the other hand, it demands that the forager makes complex decisions that integrate the state of her own crop with colony-level information (*Figure 6b*). Another possibility rids the foragers of the need to use colony level information. In this case, foragers effectively modulate their decisions to exit according to their unloading rates. We suggest a biologically appealing mechanism to achieve this, in which decisions occur at

**Table 1.** Logistic fits for a forager's probability to exit ($P$) as a function of her crop load and the colony state.

A two-dimensional logistic function of the form $P = (1 + e^{-(\alpha + \beta \cdot crop + \gamma \cdot colony)})^{-1}$ was fit to each estimated probability to exit from *Figure 6a–c*. The effect of each factor is reflected in its fitted coefficient. Within each model, effects can be compared to one another because the values of *crop* and *colony* lay on the same scale between 0 and 1. In the constant decision rate model, all coefficients were comparable in value, indicating that *crop* and *colony* had similar meaningful effects on the probability to exit. In the model where decision rate was matched to interaction rate, the effect of *colony* was weaker than the effect of *crop*, but both were still meaningful. In the model where decision rate was matched to unloading rate, the effect of *colony* approached 0 and was very weak compared to the effect of *crop*.

| Decision rate model | Factor | Coefficient | 95% CI | | $R^2$ |
|---|---|---|---|---|---|
| Constant | Intercept | $\alpha = -1.914$ | −1.921 | −1.906 | 0.81 |
| | Colony State | $\beta = -1.763$ | −1.778 | −1.747 | |
| | Forager's Crop | $\gamma = -2.330$ | −2.347 | −2.312 | |
| Interaction Rate | Intercept | $\alpha = 2.701$ | 2.686 | 2.716 | 0.92 |
| | Colony State | $\beta = -2.199$ | −2.218 | −2.181 | |
| | Forager's Crop | $\gamma = -5.810$ | −5.837 | −5.783 | |
| Unloading Rate | Intercept | $\alpha = -0.093$ | −0.105 | −0.081 | 0.71 |
| | Colony State | $\beta = 0.499$ | 0.482 | 0.515 | |
| | Forager's Crop | $\gamma = -3.092$ | −3.115 | −3.069 | |

DOI: https://doi.org/10.7554/eLife.31730.027

constant crop intervals (*Figure 6c*). Generally, any mechanism by which the forager's trigger to leave the nest depends on her unloading rate could yield similar results.

Leaving the nest while partially loaded could hold some benefits: foragers may use the food in their crops as provisions to be consumed in their expedition (*Rytter and Shik, 2016*), waiting for full unloading in the nest may be time-consuming and limit exploration for other food types, and frequent visits to the food source may ensure its exploitation. This raises the question whether there exists an optimal crop load with which foragers should exit the nest, which could potentially depend on factors such as the abundance and quality of the food source, predation risk, and the demand for food in the nest (*Dornhaus and Chittka, 2004*).

In light of our findings on both the forager's decision to exit and the distribution of interaction volumes, we hypothesize that an internal mechanism based on the mechanical tension of the crop's walls is involved in trophallaxis. Considering the crop as an elastic organ that stretches as it fills, the relative change in the volume of the recipient's crop may provide a mechanism for the scaling of interaction volumes with available crop space. Additionally, if ants could sense changes in the tension of their crop walls (*Stoffolano and Haselton, 2013*), then this would provide an anatomical basis for a model in which foragers adjust decision rates to unloading rates.

## Materials and methods

### Study species: *Camponotus sanctus*

*Camponotus sanctus* are omnivorous ants that are presumed to naturally live in monogynous colonies of tens to hundreds of individuals (projecting from *Camponotus socius*, [*Tschinkel, 2005*]), distributed from the near East to Iran and Afghanistan (*Ionescu-Hirsch, 2009*). Workers of this species are relatively large (0.8–1.6 cm) and characterized by translucent gasters, rendering them suitable for both barcode labeling and crop imaging. Our experiments were conducted on lab colonies of 50–100 workers, reared from single queens that were collected during nuptial flights in Neve Shalom and Rehovot, Israel. *Table 2* contains further details on each experimental colony.

**Table 2.** Experimental colonies.

| Experiment type | Colony | # Ants | Starvation period | # Major Workers | # Foragers |
|---|---|---|---|---|---|
| Observation | A | 100 | 3 weeks | 1 | 5 |
| | B | 62 | 5 weeks | 1 | 3 |
| | C | 53 | 4 weeks | 2 | 4 |
| Manipulation | M1 | 69 | 3 weeks | 1 | 23 |
| | M2 | 95 | 3 weeks | 2 | 16 |

DOI: https://doi.org/10.7554/eLife.31730.028

## Experimental setup

Fluorescent food imaging and 2D barcode identification (BugTag, Robiotec) were used to obtain a live visualization of the food flow through colonies of individually tagged ants. See (*Greenwald et al., 2015*) for a detailed description of the experimental setup. In short, an artificial nest was placed on a glass platform positioned between two cameras. A camera below the nest filmed through the platform, capturing the fluorescence emitted from the food inside the translucent ants. Meanwhile, a camera above the nest filmed through its infrared shelter, capturing the barcodes on the ants' thoraxes, allowing identification of single ants inside the nest. Together, footages from both cameras enabled the association between each individual ant and her food load, throughout time and across trophallactic events. The two cameras were synchronously triggered at a fixed frame rate, (here 0.5 Hz., except for colony B which was recorded at 1 Hz.). We chose a temporal resolution that is sufficient to capture events of 2 s since shorter interactions barely involve food exchange (*Greenwald et al., 2015*).

## Image processing

Top camera images were used to extract ant identities, coordinates and orientations using the Bug-Tag software (Robiotec). Bottom camera images were used to detect fluorescence with a pixel intensity threshold, using the openCV library in Python. Gasters of fed ants appeared as bright 'blobs' and thus passed the image threshold (for details, see [*Greenwald et al., 2015*]).

In order to associate between the identity of an ant and her appropriate blob, the image from the upper camera was transformed to align with the fluorescent image. Then, for each identified tag, a small area extended from the back of the tag toward the ant's abdomen was crossed with the thresholded fluorescent image. If a blob intercepted this area, it was assigned to the tag's identity.

Thus, for each experiment a database was obtained, which included for every frame the coordinates, orientation, and measured fluorescence (in arbitrary units of pixel intensity) of each identified ant.

## Interaction identification and crop load estimation

Even though the fluorescence emitted from an ant's crop is reasonably indicative of the food volume, it is a noisy measurement mainly due to her highly variable postures. Therefore, assuming that an ant's crop content remains constant during the intervals between trophallactic events, it is best evaluated as the maximal fluorescence measurement acquired in each such interval (*Greenwald et al., 2015*).

In order to precisely consider the relevant intervals for this estimation, the trophallactic interactions were manually identified from the video. Interactions were classified as trophallactic events whenever the mandibles of the participating ants came in contact and the mandibles of at least one of the ants were open. For forager ants, another situation in which their crop loads may change is when they directly feed from the food source. These feedings were also manually identified from the video, as times when a forager's open mandibles touched the food source.

Ultimately, for each ant we obtained a 'timeline', describing at every instance whether she was engaged in trophallaxis (and if so, with whom), whether she was directly feeding from the food source, and the estimated food load in her crop. *Figure 4a* depicts an example of such individual-level data.

## Observation experiment: Monitoring food flow as hungry colonies gradually satiate

Following a food-deprivation period of 3–5 weeks, ant colonies (queen, workers and brood) were manually barcoded and introduced to the experimental nest for an acclimatization period of at least 4 hr. The nest consisted of an IR-sheltered chamber (~100 cm$^2$), neighboring an open area which served as a yard (*Figure 1a*). After the acclimatization period, the two cameras synchronously started to record. After 30 min, the fluorescent food (sucrose [80 g/l], Rhodamine B [0.08 g/l]) was introduced to the nest yard *ad libitum*, and the recording proceeded for at least four more hours - a duration sufficient for the colony to reach its desired food volume intake (*Figure 1b* and *Figure 1*).

Overall, we analyzed data from three such experiments, that included 12 foragers, who fed from the food source 139 times, and were engaged in 1227 trophallactic interactions.

## Perturbation experiment: introducing hungry ants after initial satiation

To manipulatively examine the role of the colony's satiety in the control of food inflow, we characterized the system's response to a perturbation in the colony's satiety level. This experiment was conducted as the observation experiment described above, except that it consisted of two phases:

*Phase 1*: The starved colony was segregated between two equally-sized chambers - one with access to the nest yard, and the other blocked behind a removable perspex wall. Thus, when the fluorescent food was introduced to the nest yard, only the ants with access to the yard gradually satiated while the others in the blocked chamber remained hungry. We reasoned that if foragers react to the colony's satiety through their experience in the nest, they would perceive saturation of the accessible chamber as saturation of the colony, as they could only interact with ants of the accessible chamber.

*Phase 2*: After the first chamber satiated, we introduced the hungry ants of the blocked chamber by removing the wall, effectively dropping the perceived satiety level of the colony at once. Recording then proceeded for at least 90 more minutes, sufficient for the colony to reach secondary satiation (*Figure 4d* and *Figure 4—figure supplement 2*).

Segregating the colony into two chambers. In order to avoid artificial biases in the chambers' populations, ants were initially introduced to the nest without the wall to freely settle within it. Only after a habituation period of at least 4 hr, the wall was gently inserted to divide the ants, that were then left to habituate for at least one more hour before recording started. The blocked chamber included the queen and brood in both perturbation colonies, and the number of ants in the accessible chamber was 33 and 31 in colonies M1 and M2, respectively.

Time of wall removal. Satiation of the first chamber was identified with semi-online approximative image analysis of the videos from the fluorescence camera, by summing the pixel intensities of each frame, which rose as food accumulated. Satiation was determined when this fluorescent signal ceased to rise for at least 1 hr, serving as our cue to remove the wall.

Overall, we analyzed data from two perturbation experiments, on colonies of 69 and 95 ants, including 23 and 16 foragers, respectively.

## Data analysis

All data obtained after crop load estimation was analyzed using Matlab software. Four data files are available with this manuscript.

## Foragers

Each experiment consisted of a few individuals who performed consistent foraging cycles between the food source and the nest. Those ants were considered as 'foragers'. Some other individuals were occasionally observed at the food source but clearly did not display such foraging cycles. To our purposes they were not considered as foragers. These ants visited the food source no more than four times, while consistent foragers performed an average of 15.67 cycles and no less than 8. The data presented here is from the first return of a forger to the nest from the food source until the end of the experiment.

## Food flow

The total accumulated food was calculated as the sum of all interaction volumes between foragers and non-foragers. The volume of an interaction was taken to be positive when food was transferred from the forager and negative when it was transferred to the forager. Each forager's contribution is the sum of her own interaction volumes. Although food accumulated through discrete local events, we were mostly interested in the average dynamics of food flow, which are convenient to describe in a continuous manner. Therefore, the accumulated food was first smoothed with a moving average with a time window large enough to include several trophallactic events (2000 s). This window size was chosen by plotting the smoothed data on top of the raw data and assuring that small fluctuations were smoothed while the general shape was maintained. Food inflow was derived by differentiating the smoothed data. Since differentiation is a process highly sensitive to local noise, we differentiated the smoothed accumulated food with a window of 200–500 s, depending on the fluctuations of the experiment. This window size was chosen by verifying that the sum of the obtained inflow is indeed sufficiently close to the raw data of accumulated food.

## Food load normalization

Our experimental method provided us with measurements of food volume in arbitrary units of fluorescent pixel intensity. Due to possible variations in lighting conditions between experiments, the obtained pixel intensities were incomparable. Therefore we used pixel intensities only for analyses performed within the same experiment (*Figure 1b–c*, *Figure 4d–e* and *Figure 1—figure supplement 1*). For all other purposes, food load was estimated in normalized units. In analyses where individual crop loads and interaction volumes were linked to the global dynamics (*Figure 3*), absolute loads were important. Therefore, food loads were normalized between experiments, by dividing each measurement by the $90^{th}$ percentile crop load measurement of its experiment. In analyzing foragers' responses to their own crop loads (*Figure 6*), the relative satiety state of each forager was of interest. Accordingly, food loads were normalized between foragers, by dividing the measurements of each forager by her own maximal measurement.

## Exponential fits to interaction volume distributions

Since exponential distributions could be fit only to 'positive' interactions, i.e. where the forager was the donor, when we fit exponential distributions we neglected the negative interactions. Negative interactions constituted 216 out of 962, and accounted for 12% of the total food flow. The consequence of this approximation is that we effectively lose 12% accuracy in the modeled food flow. Despite this loss of accuracy, the results from this analysis were consistent with parameters obtained otherwise (without neglecting the negative interactions), ensuring that it was indeed sufficient to consider only the positive interactions.

## Acknowledgements

We would like to acknowledge E Segre, G Han, Y Burnishev for the technical help and Y Dover and R Harpaz for statistical advice. We also thank E Fonio, Y Heyman, A Gelblum, H Rajendran, A Le Boeuf and T Halperin for instructive comments on the manuscript. OF is the incumbent of the Shloimo and Michla Tomarin Career Development Chair, was supported by the Israeli Science Foundation grant 833/15 and would like to thank the Clore Foundation for their ongoing generosity. This research is further supported by a research grant from the Estate of Rachmiel Ramon Bloch and by the European Research Council (ERC) under the European Unions Horizon 2020 research and innovation program (grant agreement No 648032).

## Additional information

### Funding

| Funder | Grant reference number | Author |
| --- | --- | --- |
| Israel Science Foundation | 833/15 | Ofer Feinerman |
| European Research Council | 648032 | Ofer Feinerman |

| The Estate of Rachmiel Ramon Bloch | Ofer Feinerman |
| --- | --- |
| Clore Duffield Foundation | Ofer Feinerman |

The funders had no role in study design, data collection and interpretation, or the decision to submit the work for publication.

### Author contributions

Efrat Esther Greenwald, Lior Baltiansky, Conceptualization, Software, Formal analysis, Writing—original draft, Writing—review and editing; Ofer Feinerman, Conceptualization, Supervision, Funding acquisition, Writing—original draft, Writing—review and editing

### Author ORCIDs

Efrat Esther Greenwald http://orcid.org/0000-0003-3614-8915
Lior Baltiansky http://orcid.org/0000-0003-3870-1788
Ofer Feinerman http://orcid.org/0000-0003-4145-0238

### Decision letter and Author response

Decision letter https://doi.org/10.7554/eLife.31730.031
Author response https://doi.org/10.7554/eLife.31730.032

## Additional files

### Supplementary files

• Transparent reporting form
DOI: https://doi.org/10.7554/eLife.31730.029

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
