## [Decision Letter]

Thank you for submitting your article "Individual crop loads provide local control for collective food intake in ant colonies" for consideration by *eLife*. Your article has been reviewed by two peer reviewers, and the evaluation has been overseen by a Reviewing Editor and Ian Baldwin as the Senior Editor. One of the two reviewers, Danielle Mersch, has agreed to reveal her identity.

The reviewers have discussed the reviews with one another and the Reviewing Editor has drafted this decision to help you prepare a revised submission.

Summary:

This paper employs a novel quantitative imaging technique to investigate the microscopic dynamics of trophallaxis dynamics within ant colonies. By integrating this method with a simple individual-level model the authors have demonstrated that there exist simple, but effective, feedback mechanisms whereby colony-level nutritional regulation can be mediated by individuals that react only to their own satiation levels.

Essential revisions:

1) This work needs to better take into account previous studies in this area. Notably the highly-relevant studies on ants (i.e. Traniello, 1977; Buffin et al., 2009) and honey bees (i.e. Seeley, 1989) must be cited, and the present findings discussed in the light to these previous works. You should clearly balance explanations of how your results – and the application of a new method – represent a significant advance in the field by describing precisely how their work builds upon the prior research cited above and significantly advances the field.

2) Regarding data binning in Figure 2, Figure 3, Figure 4, Figure 5, Figure 2—figure supplement 2, Figure 3—figure supplement 2, Figure 4—figure supplement 1 - the data are binned without explanation being given regarding the number and size of bins that were chosen. In addition, the number of bins varies between plots based on the same variable; see e.g. Figure 2, Figure 2, Figure 2 all depict colony state, but Figure 2 and Figure 2 have 5 bins, while Figure 2 has 8 bins. Because data binning can alter the shape of lines and curves, the choice of binning gives the impression that it may have been adjusted to produce 'desired' results. Please conduct a sensitivity analysis of the binning methodology, and/or refrain from binning and plot all data points instead (like in Figure 2—figure supplement 4), ideally color-coding them by forager ID, and/or colony ID to provide a more comprehensive view to the reader.

3) It was highlighted that sometimes the reader could fail in grasping what is meant by some statements (e.g. Subsections “Global feedback from local interactions” and “Foraging Effort is Matched to the Colony's Need”). Care should be given to the clarity of writing and grammatical correctness in the revised manuscript to ensure that the messages come across clearly. Also, please do not rely on mathematical expressions alone to make a point; please spell out the messages in plain writing as well as providing the equations.

4) It is presently unclear to how the input data used in the analysis, and the associated scripts, will be made available. The database mentioned (Subsection “Image processing”) and all scripts used to run their analysis must be made available. Information on the details of the analyses are relatively limited and require searching through Figures and the Material and methods section. Too little detail is given on the criteria used to exclude data points, on the number of points/foragers excluded, on how the size of the smoothing windows were selected, or which algorithms were used, and why. Similarly, as noted above, there is no information on the logic behind the data binning, nor on how many data points were averaged for each point in their plots. Details on replicates are present but hidden in the material and methods section. The statistical methods are also too unclear, particularly with respect to whether tests were done per colony, and p-values combined, or whether colony ID was included as random factor, or completely ignored. Individual data are grouped as averages, and maybe even across colonies, so the reader cannot evaluate the extent and type of variability that exists within the data. These impact the statistical validity of the work. Furthermore, the reader must be able to replicate these analyses, and the appropriate information must be given in order to do so.

---

## [Author Response]

We would like to begin by thanking the referees for the comprehensive review. We have edited our manuscript accordingly. Major edits include an extensive revision of the Introduction, a better explanation of all mathematical progressions, and splitting Figure 2 into two figures (Figure 2 and Figure 3) such that the new Figure 2 (where new information was added) provides a direct comparison showing where our work diverges from previous works. We believe that this revision better places our paper in context of previous knowledge as well as significantly improves readability.

Essential revisions:1) This work needs to better take into account previous studies in this area. Notably the highly-relevant studies on ants (i.e. Traniello, 1977; Buffin et al., 2009) and honey bees (i.e. Seeley, 1989) must be cited, and the present findings discussed in the light to these previous works. You should clearly balance explanations of how your results – and the application of a new method – represent a significant advance in the field by describing precisely how their work builds upon the prior research cited above and significantly advances the field.

This comment seems to imply that all three papers were not cited in the original version of this paper. Since *eLife* publishes these comments and their answers with the manuscript we would like to set the record straight and state that Buffin et al., 2009 and Seeley, 1989 were cited and, more than briefly, discussed in the first version of this manuscript. We thank the editor and referees for pointing us to Traniello’s impressive work from 1977 (more about this work in our answers to the specific points below).

We have now extensively revised the Introduction of this paper so it more clearly states the general framework, the specific study question, what was previously known, how our work builds on this knowledge and what are the caveats in the earlier knowledge that we attempt to narrow using our new technological tools. The Introduction relies on 30 citations of previous work. The new first paragraph of the Discussion section now explicitly states our two, main new micro-scale measurements that deviate from what was previously assumed. The following two subsections of the Discussion section discuss the implications of all our findings by comparing them with the existing knowledge in a detailed point by point manner (including the relevant citations) and discuss those aspects in which previous understanding of food accumulation in ant colonies must be expanded or revised.

The main contributions of this work are:

A) A fully quantitative model for collective inflow from local rules. Previous works of food flow regulation provided mostly qualitative intuition for the connections between the micro and the macro of this system. In this work, we provide a detailed quantitative model and quantitatively verify it by relying on microscopic scale measurements (detailed in the Introduction) that were never before performed. The difference between our quantitative model and older qualitative explanations is now expressed the Discussion section.

B) New empirical data on the microscopic scale and its implications. Our new microscopic-level data provides quantitative support for some previously assumed notions while quantitatively contradicting others. These contradictions differ in essence from previously assumed notions (Introduction), and lead to questions that have not been addressed or even raised before.

For example, a core contradiction that our data reveals, is that a typical trophallactic interaction does not fill a recipient up to her capacity, even if the forager has enough food in her crop to do so. This pulls the rug from under the individual-level reasoning that was previously used to explain various aspects of colony food intake, such as: a forager's unloading rate in the nest (Seeley, 1989, Buffin, 2009, Sendova-Franks, 2010), the number of trophallactic interactions in which she participates (Huang, 2003, Gregson, 2003), the distribution of crop loads in the colony (Gregson, 2003), and perhaps more. This finding should shift the attention from the factor that was previously focused on, the rate of trophallactic events, to the previously neglected factor, the amount of food transferred in interactions (see Figure 2 now expanded and moved from the Supplementary information to the main-text and modified to reflect this, subsection “Global feedback from local interactions” and subsection “Regulation of food flow”). While our work draws a quite comprehensive quantitative account for the links between the distribution of interaction volumes and the regulation of collective food flow, new questions are now called for: what causes the characteristic exponential distribution of interaction volumes? If it is not physical constraints of crop capacity that determine interaction volumes, what does? And why is it that these distributions appear when a simple all-or-nothing distribution could have yielded similar collective results? These new notions are discussed in subsection “Regulation of food flow” and subsection “Regulation of foraging trips”.

The second core contradiction we found was that foragers do not unload their entire crop contents before leaving the nest to reload at the food source. Similarly, to the above contradiction, this unintuitive finding means that previous reasoning regarding the number of trophallactic interactions of a forager (Gregson, 2003), the distribution of crop loads in the colony (Gregson, 2003), and the frequency of a forager's exits (Buffin, 2009), is wrong. The new questions that rise from this observation have to do with the timing of foragers' exits: If a forager's trigger to re-exit the nest is not the complete emptying of her crop (and not even an obvious crop load threshold), what is it that triggers a forager to leave? What kind of computational and sensory abilities are demanded from a forager in order to generate regulation of foraging frequency? How does the average crop load with which a forager exits remain constant at different colony state conditions, even though there is a positive probability to leave the nest at a wide range of crop loads, and different colony states provide different prior distributions of foragers' crop loads? These questions are addressed in our analyses (Figure 6), which introduce a suggestion for a simple biologically-plausible mechanism for regulation of foraging frequency with a wide range of crop loads and few computational demands, which has not been described before (Discussion section). A new question that follows is: why do foragers leave the nest with a wide range of crop loads even though it seems to be energetically wasteful and make it harder to regulate foraging frequencies? This notion is discussed in subsection “Regulation of foraging trips”.

C) Data sharing. The novel microscale data on real-time individual crop loads and trophallactic food transfer, which is made fully available as Source data files that accompany this submission, holds value on its own. It provides a quantitative glimpse for those interested in ant nutrition, distributed regulation, and social processes, into an otherwise unseen natural distributed process.

2) Regarding data binning in Figure 2, Figure 3, Figure 4, Figure 5, Figure 2—figure supplement 2, Figure 3—figure supplement 2, Figure 4—figure supplement 1 - the data are binned without explanation being given regarding the number and size of bins that were chosen. In addition, the number of bins varies between plots based on the same variable; see e.g. Figure 2, Figure 2, Figure 2 all depict colony state, but Figure 2 and Figure 2 have 5 bins, while Figure 2 has 8 bins. Because data binning can alter the shape of lines and curves, the choice of binning gives the impression that it may have been adjusted to produce 'desired' results. Please conduct a sensitivity analysis of the binning methodology, and/or refrain from binning and plot all data points instead (like in Figure 2—figure supplement 4), ideally color-coding them by forager ID, and/or colony ID to provide a more comprehensive view to the reader.

We chose to present binned data for two reasons: First, for visual clarity – the trends are most pronounced in the average values, which we obtain by binning. The average values are also the system parameters in our analytical framework. Second, our dataset has an inherent bias, by which due to the dynamics of food accumulation, there are more data points in high colony states compared to low colony states. Binning the data balances the relative weight across colony states, despite the bias in the number of datapoints.

That being said, we acknowledge that binning may be a sensitive procedure, and that raw data is valuable on its own. We therefore supplement our binned plots with plots containing raw data. Additionally, results of sensitivity analysis on binning methodology are now added, where relevant, to the Supplementary material. More detailed information on the number of points averaged in each bin is now provided in the figure captions. Figure 5 is modified to include a color-coding scheme by forager ID.

Binned figures:

Figure 2 (interaction rate) – In addition to plotting the raw data (Figure 2—figure supplement 1), since this figure presents a fit to the averaged bins we made sure that different binning choices do not affect the obtained fit. To that end we performed a series of fits for a range of bin numbers from 4 to 10 bins. The results of each fit are presented in a table in the Supplementary Information (Table S1). Indeed, binning did not substantially affect the fit, as can be seen by the obtained values and their small STD compared to their mean. Originally, what we concluded from the fit was that the change in mean interaction rates is much less prominent than the change in mean interaction volumes. Importantly, this conclusion holds for any of the fits obtained from the series of varying bin numbers: in all interaction rate fits, the slope is close to 2-folds smaller than the intercept, as opposed to interaction volumes – for which both values are the same.

Figure 2 (interaction volume) – Here we present raw data alongside the binned data in the Supplementary information (Figure 2—figure supplement 2). The red line in the plot Figure 2 is not a fit, but a prediction from our analyses that has nothing to do with the specific choice of bins. Additionally, the prediction is about mean interaction volumes, such that the bin averages are the relevant parameter to compare to. To show that this representation is not misleading due to the specific choice of bin number, we fit a linear function to both raw data and binned data – in both cases the slope and the intercept were indeed extremely close to each other, and moreover – close to the predicted value. This data is now also shown in the caption of Figure 2—figure supplement 2.

Figure 3 (previously Figure 2) – These figures were meant to show that the recipient’s crop load affects the distribution of interaction volumes, while the forager’s crop load does not. This conclusion was obtained also by another method that is less bin-sensitive and which was previously included in the supplementary. To avoid presentation of visually misleading information, we chose to move the supplementary results to the main text instead of the previous plots. They now include the mean and the STD of the parameter 𝜆_c_ as obtained from various fits of the same form that were applied to a series of 17 bin size choices uniformly covering the range [0.01-0.09]. This procedure is explained in the caption of Figure 3, and fit examples are Figure 2—figure supplement 4 and Figure 3—figure supplement 1).

Figure 3 (previously Figure 2) – Apart from binned averages, this panel also includes distributions of the full data, which we believe is the best way to evaluate the variation while allowing to compare between recipients’ states and the general state in the colony. Therefore, we chose to leave this plot as is, while the raw data is provided in a data source file. We now specify in the caption the number of datapoints in each bin: The blue bins include n =84, 137, 165, 274, 496 data points for bins 1-5, respectively and the red bins include n=202 data points per colony state bin.

Figure 4 (previously Figure 3) – Are now accompanied by the full data shown in Figure 3—figure supplement 3. To ensure that the fit in Figure 4 is not sensitive to the chosen bins, we performed it also on the raw data – both procedures yielded the same fitted line (Figure 3—figure supplement 3). This is explained in the caption of Figure 3—figure supplement 3. Since Figure 4 presents the same data as in Figure 4, and since no quantitative fits were made to it, we did not separately evaluate its binning – we present its raw data in Figure 3—figure supplement 3. The number of data points in each bin is now specified in the caption of the Figure: n=26,26,28,39,57, for bins 1-5, respectively.

Figure 4 (previously Figure 3) – No quantitative fit was made to the data presented in this figure, it just visually shows the response to the manipulation. Therefore, its sensitivity to binning is not quantified, but the raw data is presented in Figure 4—figure supplement 1. The number of data points within each bin are now specified in the Figure caption: n =28,36,21,14,9,28,27,19,5, for bins 1-9, respectively.

Figure 5 (previously Figure 4) – This figure includes now the raw data, and the number of data points included in each bin is mentioned in the caption: n=11,15,13,13,15,13,17,22,27,30 for bins 1-10, respectively. The claim that the foragers’ crop loads at exit are not strongly dependent on the colony state is supported by Spearman’s correlation test that was performed on the raw data (subsection “Foragers' crop loads upon leaving the nest”), such that the specific choice of bins does not affect it.

Figure 5 (previously Figure 4) – This Figure was modified to include a color-coding by forager ID. The data is available in the source data file.

Figure 5 (previously Figure 4) – As in Figure 4, no quantitative fit was made to the data presented in these plots, so their sensitivity to binning was not quantified, but raw data is shown in Figure 4—figure supplement 2. The number of data points within each bin is now specified in the figure caption: n=26,26,28,39,57 for bins 1-5, respectively.

Figure 6 (previously Figure 5) – The statistical support for the results presented in Figure 6 was obtained with a method that is more appropriate to the structure of our data. The GLM analysis that was used in the previous version was done on raw binary data (exit vs. stay), which was not homogeneously distributed over our space parameters. Since GLM was shown to be sensitive to biases in data distribution, the fitted model could have been misleading. In this version of the manuscript we overcame this bias by fitting the same logistic function as used in the GLM to the binned data which evenly spreads the weight of the data on the parameter space (see Table S2). This method is not sensitive to the binning procedure, since bins are equally-sized and highly overlapping, with the binned value located at a location that takes into account the distribution of data-points within the bin.

Figure 6 (previously Figure 5) – The GLM fit presented in this panel was obtained by using the binary data presented in panel 5C. The plotted data points represent fractions of “exit/stay” calculated from bins of the binary data, which were chosen by Matlab’s default ‘auto’ algorithm, which chooses a bin width to cover the data range and reveal the shape of the underlying distribution. Those points are for visualization only, they must come from binned data by nature, and they do not affect the GLM. Here the GLM does not suffer from the weakness explained above in Figure 6, since this is a one-dimensional GLM for the data that is distributed only over the forager’s crop load factor. Over this dimension the distribution of the data is sufficient for a reliable GLM.

Figure 2—figure supplement 2 – see Figure 2.

Supplementary Figure 5 is now presented as Figure 3 in the main text. For details see Figure 3 above.

Supplementary Figure 5 is now Figure 3—figure supplement 3 (‘Mean Interaction Volumes as a Function of the Recipient’s Crop Load’) in the new version of the manuscript. It includes the raw data. The same fit was obtained for different binnings (5-10 bins), such that we ensure that it is not sensitive to binning. This is explained in the figure caption. The number of data points in each bin (n=26,26,28,39,57 for bins 1-5, respectively) and the fitting result for different binning are specified in the caption.

Supplementary Figure 6 is Figure 4—figure supplement 2 (‘Perturbation Experiments’) in the new version of the manuscript. It includes the raw data and the number of data points in each bin are specified in the figure caption (M1:

n=47,13,15,13,12,16,17,14 for bins 1-8, respectively; M2: n=28,36,21,14,5,15,18,10,4, for bins 1-9, respectively) and those of foragers that began foraging after the manipulation (M1: n=6,3,2, for bins 1-3, respectively; M2: n=4,13,9,9,1, for bins 1-5, respectively).

3) It was highlighted that sometimes the reader could fail in grasping what is meant by some statements (e.g. Subsections “Global feedback from local interactions” and “Foraging E_ort is Matched to the Colony's Need”). Care should be given to the clarity of writing and grammatical correctness in the revised manuscript to ensure that the messages come across clearly. Also, please do not rely on mathematical expressions alone to make a point; please spell out the messages in plain writing as well as providing the equations.

The entire manuscript was extensively edited for clarity and grammar. Changes were done in Results section so that all mathematical expressions are also explained in words. In general, the subsection titled “Global feedback from local interaction” has been significantly edited. We have broken down the mathematical reasoning into easy-to-read steps all of which are explained at length within the text.

4) It is presently unclear to how the input data used in the analysis, and the associated scripts, will be made available. The database mentioned (Subsection “Image processing”) and all scripts used to run their analysis must be made available.

We have now organized all the data in Excel files, and MATLAB executable. As these are not too large, we have attached these files as Source Data files along with the paper.

We note here that while doing this we have found a normalization error in the definition of the colony state of one of the three analyzed colonies. The error was small, and its correction had no significant impact other than small quantitative corrections from our previous version. This had no effect on any of our qualitative findings or conclusions. We apologize for this confusion.

Information on the details of the analyses are relatively limited and require searching through Figures and the Material and methods section.

The information about the analysis has now been elaborated. References to the relevant place in the Supplementary Information or the Materials and methods section were carefully made within each figure caption.

Too little detail is given on the criteria used to exclude data points, on the number of points/foragers excluded, on how the size of the smoothing windows were selected, or which algorithms were used, and why.

In this version we elaborate more on these issues.

Exclusion of data points:

The data presented here is from the first return of a forger to the nest from the food source until the end of the experiment. We generally did not exclude data points, unless otherwise specified as in Figure 6 (see the caption of Figure 6) and Figure 3 (see Supplementary Information 2.4 ‘Fitting an exponential function to the distribution of interaction volumes’).

In the subfigures of Figure 3, only interactions in which the forager was the donor (81% of interactions, subsection’ Exponential fits to interaction volume distributions’) were considered, since we modeled the inflow of food into the colony. In addition, the exponential distributions of interaction volumes that we obtained, support that negative interactions (i.e. where the forager is the recipient), which cannot distribute exponentially, are not part of the process in interest.

In Figure 6, observation points during trophallaxis were excluded, since being engaged in trophallaxis prevents the possibility to exit the nest in the next observation (after 2 seconds).

In Figure 6, since foragers' crop loads rarely rose in the nest, their lowest crop observation in a visit was generally an `exit', so the calculated probability to exit in the lowest crop interval was 1. To ensure that the crop state played a role beyond this extreme effect, we fit a GLM excluding the lowest crop interval (Figure 6 caption).

Smoothing:

The single place where smoothing was performed was in the panel describing food accumulation data (Figure 1 and Figure 1—figure supplement 1). The lines presented here are smoothed versions of the raw data that use a simple moving average filter. Window size was chosen by comparing smoothed data to the raw data. We now include the smoothed data along with the raw data to show that the smoothed data indeed captures the original shape (Figure 1—figure supplement 1 in the SI and ‘Food flow’ subsection in the Materials and Methods section.

Similarly, as noted above, there is no information on the logic behind the data binning, nor on how many data points were averaged for each point in their plots.

This point was addressed in the answer to comment 2 above.

Details on replicates are present but hidden in the material and methods section.

Replicate information is now easier to find, aggregated in ‘Table 1: Experimental Colonies’.

The statistical methods are also too unclear, particularly with respect to whether tests were done per colony, and p-values combined, or whether colony ID was included as random factor, or completely ignored. Individual data are grouped as averages, and maybe even across colonies, so the reader cannot evaluate the extent and type of variability that exists within the data. These impact the statistical validity of the work.

All statistical analysis were performed on the pooled data from all colonies and foragers with no grouping. Neither colony ID nor forager ID were considered as random factors. Variability of the data can now be evaluated from the supplementary figures where the raw data is included. This information is now included in the captions of Figure 2, Figure 3, Figure 4, Figure 5, Figure 6, Figure 1—figure supplement 2, Figure 1—figure supplement 1, Figure 2—figure supplement 2, Figure 2—figure supplement 3, Figure 3—figure supplement 1, Figure 3—figure supplement 2, Figure 3—figure supplement 3, Figure 4—figure supplement 1 and Figure 5—figure supplement 1.

Furthermore, the reader must be able to replicate these analyses, and the appropriate information must be given in order to do so.

We agree with this comment and have made our explanations more detailed. As noted above this detailed information about the analyses is now present in the caption and supplementary information of the figures. The attachment of Source data files to this submission should also be of help in this respect.